# Relationship between Corporate Sustainability and Compliance with State-Owned Enterprises in Central-Europe: A Case Study from Hungary

**Anita Boros [1] and Csaba Fogarassy [2],*** 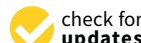

[1]   Center for Globalization Competences, Széchenyi István University, University Square 1,
      9026 Győr, Hungary; anita.boros@itm.gov.hu
[2]   Climate Change Economics Research Centre, Szent Istvan University, Pater Karoly 1, 2100 Godollo, Hungary
*    Correspondence: fogarassy.csaba@gtk.szie.hu

**Abstract:** Our study deals with the control of Hungarian state-owned business associations in order to find out whether there is any correlation between corporate sustainability and compliance. According to our hypothesis, the state has a greater responsibility for the sustainable operation of state-owned enterprises—one of the tools of which can be the efficient construction of so-called compliance controls. A state-owned enterprise can be sustained in its operation and function by doing what it has been assigned to it as a task. The sustainable operation can be achieved through the use of circular feedback and continuous control. Corporate sustainability can be influenced by a number of factors that are crucial to the integrity and adequacy of companies. In our study, these are the Initial Hazard Factors (IHFs), Hazardous Increasing Factors (HIFs), and Control Enhancement Factors (CEFs), which were used in indexed form for testing. For the specific analysis, we used the Analytical Hierarchy Process (AHP) method to rank and evaluate risk avoidance options. We analyzed the practice of the State Audit Office of Hungary and its results and found that in the case of state-owned enterprises, the current Hungarian legal system does not contain uniform normative regulations that would regulate the control of conformity in a broader sense. As a result, corporate integrity and corporate compliance are also subject to a fragmented set of rules in different jurisdictions. This has a negative impact on the development of the company's long-term, sustainable operating principles. Based on our research, a number of factors have been identified (including enterprise size and intensive use of EU funding resources) that may result in a loss of corporate sustainability but can be mitigated or even eliminated by creating an effective internal control environment. Based on literature analysis, most of the Hungarian phenomena are typical of the former socialist countries of Central Europe. The interpretation of sustainability—in transition management countries—for first-generation enterprises which were developed in a socialist market economy is quite different from the most developed countries of the European Union. The main reason for this is that generational rules do not yet exist.

**Keywords:** sustainability deficit; corporate sustainability; compliance; integrity; control systems; state-owned enterprises; Analytical Hierarchy Process (AHP)

---

## 1. Introduction

Corporate sustainability is a complex system [1]—a strategic and operational process for the entire corporate structure that creates long-term values for the organization and society as a whole. At the same time, corporate sustainability goes beyond the general guidelines of sustainability [2], as most sustainability standards tend to provide minimal guidance for companies to develop a

sustainable corporate model [3]. Undoubtedly, corporate sustainability has long been a focus of interest for prominent representatives of literature, but its unified conceptual definition has not been found. In the corporate literature, we see a widely differing view of corporate sustainability criteria, which are often radically opposed to one another. Trends are often assigned to one author, so their identification cannot be based on consistently definable criteria. Without completeness, we emphasize that Springett 2003 [4] primarily builds on the self-esteem of multinational companies [5], and economic approaches appear in absolute and relative terms: the most important benefits of a given company in the absolute approach to deducing the costs of enterprise activity, in order to get to know the sustainability impact of the company. This approach is the application of two sustainability indicators at the national level related to neoclassical economics theory at the company level. The relative approach, on the other hand, provides the benefits of one unit of environmental or social impact at the enterprise level [6,7]. According to the heuristic approach, corporate sustainability has to meet five interrelated challenges—economic efficiency, social effectiveness, and social efficiency and integration [8]. The eco-centered management approach focuses on the interdisciplinary exploratory framework for a new feature of environmental issues, highlighting institutional failures in risk management in modern societies [8]. Although corporate responsibility is a curious issue, CSR Corporate Social Responsibility (CSR) is still a more popular topic in terms of the number of related literature publications [9]. However, we believe that corporate sustainability is not just a scientific issue, as there is an even more intense need for business associations to show how they contribute to the government's national sustainability goals [6,10]. This expectation is even more pronounced for state-owned enterprises, where the proper management of ecosystem services and the huge databases associated with the fulfillment of public tasks are a particular great social responsibility [11].

Operating in compliance with sustainability principles is increasingly becoming a business value, as the finality of resources concerns economic players, so they should encourage investments that provide long-term competitive advantage. That is, they must be subject to their corporate goals—sustainability criteria that have not yet been developed consistently [12]. In recent years, the circular economy, augmented by business and eco-innovation, has emerged as a synonym for sustainability, offering novel and well-described basic economic mechanisms and business solutions to pursue sustainable solutions that were not previously reported as part of everyday practice [13,14]. These include circular feedback from market players or government guarantees for social and environmental benefits. This new business solution can complement the monitoring of the resources used and the measurement of their efficiency which also depend on the company size [15,16].

Exploring the conditions for corporate sustainability is a broad, multi-scientific endeavor dealing with the mechanisms of "sustainability-driven entrepreneurship" (SDE), for which it is worth highlighting that investigations focus primarily on small and private businesses. According to the literature, the sustainable operation of state-owned or partly state-owned enterprises remains an open question. According to Sharma 2015 [17], three generalizable statements can be made that typically increase the sustainability-driven characteristic. First, he notes that self-interest or profit-making does not enhance sustainability. In the case of state-owned enterprises, this set of objectives is basically given. That is, they operate for non-profit purposes [18]. The second important aspect is the competence-driven business and the appropriate practical competence. We also investigated this in the case of state-owned enterprises, according to which this system property is not a strength of state-owned enterprises. Third, the application of generational rules is mentioned by Gibbs 2006 [19], during which the application of generational, intergenerational habits and their sustainable interpretation are key issues. The application of generational rules (e.g., resource management that strengthens locality) in the transition countries of the Union (especially in the former socialist countries) is also difficult to interpret because economic operators follow a previously unused practice [20].

The basic mechanisms for state-owned enterprises that are easily distinguishable (environmental, social, and economic) and can be defined as a corporate target in a separate package are most easily understood. However, these sustainability programs are usually not part of logical and

consistent corporate governance and can therefore be easily sacrificed due to any economic problems. These in-house relationships need to change in the future.

In terms of corporate sustainability, there are basically three dimensions: environmental, social and economic. It is now widely accepted that consideration of both environmental (including renewable energy) and social aspects is essential for the long-term survival and success of the company [21,22]. The key issue of our topic is the relationship between corporate sustainability and compliance, and what factors can affect them at the level of public companies. For companies, sustainability means replacing the previous approach, optimization based solely on economic interests, with the triple bottom line—that is, taking social, economic and environmental goals into account [23]. However, by virtue of their nature, state-owned enterprises also have to meet one of these requirements, namely the compliance requirement. As the name implies, compliance means that a company is working well towards sustainability [24]. How can checking compliance contribute to sustainable corporate governance in Hungary? Our basic hypothesis is that the state has a greater responsibility for the sustainable operation of state-owned enterprises, because these often lead to the disruption of external resources (state or EU funds) after the cessation of these resources. One of the tools to overcome this problem is the efficient construction of the newly named compliance controls. The study examines—in the Hungarian governmental practice—whether the effective use of compliance control, including circular principles, can provide a basis for the proper and sustainable performance of its functions in state-owned enterprises, in particular the size of the enterprise or the volume of the use of EU funds.

## 2. Key Aspects of Corporate Compliance

Compliance is a very complex definition or concept [25], as it includes financial, economic, tax, business, legal, ethical, sustainability, and proprietary compliance. In fact, when examining the suitability of an organization, we are looking for the answer to whether its operating mechanisms can be subordinated to all the rules, objectives, and expectations of the organization. In the narrower sense, compliance means compliance with and the enforcement of the legal regulations applicable to a business association, including the decisions of the owner. In the broader sense, when compared to positions, it means much more:

- the objectives pursued in setting up the business in question;
- ownership, sometimes at the governmental level, together with regulators at the policy (sector) level;
- compliance with standards;
- meeting the expectations of the public service users;
- compliance with the organization's short-, medium- and long-term strategic objectives;
- it also represents compliance with corporate values for managers and employees.

In the context of compliance regulation, the International Standards of Supreme Audit Institutions (ISSAI) should be highlighted. The audit principles applicable to all types of audits are outlined in the General Principles of Public Sector Auditing [26].

In this system, financial control is a control activity designed to determine whether the audited entity presents its financial information in accordance with the applicable financial reporting and regulatory framework. It has a special role in relation to public sector organizations, as it can also create public confidence in the state, an organization performing public tasks. The purpose of performance audits, on the other hand, is to determine whether the operation of the organization, its operating principles and procedures can be subordinated to the principles of economy, efficiency and effectiveness. Finally, the standard emphasizes preventive mechanisms in the context of compliance auditing and states that such auditing should be aimed at examining whether a particular management decision, measure, or regulator within an organization meets predefined compliance criteria—that is, to what extent it is possible to minimize the compliance risks already mentioned [27–31].

A separate standard has also been adopted to regulate this control function [26]. This standard emphasizes that compliance control is basically divided into two parts: first, compliance with the criteria established by regularity (compliance with formal criteria such as legal regulations, ownership decisions, or agreements) and other compliance (with criteria for sound financial management and ethical conduct [26]. It has to be emphasized here that, in itself, one of the risk factors of sustainability and thus of compliance may be that these regulations have been changing frequently [20,25].

## 3. Overview of the Compliance Environment for Hungarian State-Owned Enterprises

State-owned enterprises are in a special position in the system of public tasks: they are directly or indirectly owned by the state, using public funds and generally contributing to the performance of a public task. State-owned enterprises are basically established when the task to be performed by the state can be accomplished more effectively through a business entity owned by the state than by a budgetary organization [32].

The OECD has also examined the relationship of the state to its own companies and, in its recommendations, draws attention to the particular importance of the state for the public policy purpose of the state enterprise. Adequate ownership management mechanisms should be developed that best serve the achievement of these goals without limiting the responsibility of directors of state-owned companies to intervene in governance [33]. From the point of view of our topic, it should be emphasized that the state, besides defining the target system to be applied for the given company, must specify the tolerable level of risk, and it must properly control the achievement of these. With regard to control mechanisms, the OECD also emphasizes that the use of public funds by state-owned companies must be transparent and in line with international financial standards for private sector companies [34]. The increasing state control and the appreciation of the ownership system of the OECD member states examined by the OECD were also observed in Hungary after 2010 (OECD, 2019). The Hungarian economy has been struggling with deepening problems since the early 2000s: the public debt, low efficiency fiscal policy and weak control potential continued to deepen during the 2007–2008 crisis [35]. The change has been evident in many areas since 2010.

In the Hungarian model, based on active state involvement, the purpose of the control system at the national economy level has shifted in a direction that facilitated the actual intervention of the entitled person in the financial processes. These requirements must also apply to state-owned enterprises, supporting the requirements of good governance [36]. Expectations were formulated in a very high level of legal regulators: Article 38 (1) of the Hungarian Fundamental Law states that "State and local governments shall own property. The purpose of managing and protecting national wealth is to serve the public interest, to meet common needs and to conserve natural resources, and to address the needs of future generations" [35]. Article 5 (5) states that "State and local government-owned enterprises shall operate in the manner prescribed by law, in an autonomous and responsible manner, in accordance with the requirements of legality, expediency and effectiveness". Article 39 (2) of the Fundamental Law adds that "all organizations that manage public money are required to account for their public money management". Public money and national wealth must be managed in accordance with the principles of transparency and fairness of public life. The proper management of public money is therefore an obligation deriving from the Fundamental Law. These constitutional rules are complemented by property laws. Act CXCVI of 2011 on National Propriety in the Nvtv. states that the essential function of national wealth is to ensure the performance of public tasks, including the provision of public services to the population and the provision of the infrastructure necessary for the performance of these tasks.

National Property shall be managed in a manner that is responsible for its intended purpose. In addition to the transparency clauses contained in the Nvtv., Act CVI of 2007 on State Property states that the state may only participate in or establish an enterprise in which its liability does not exceed the amount of its financial contribution. On behalf of the state, the owner of property rights has a

duty to enforce the responsibility of the management and the management of [31] property in the public interest.

In addition to the general property law regulators, the civil law provisions defining the general rules of business operation and Act CXXII of 2009 on the more economical operation of public-owned companies play an important role (the Savings Act.), as well as its specific rules for state-owned enterprises. The sources of law laying down public finance regulations are also a decisive source, in particular the laws on public finance, accounting, taxation and certain types of taxes and levies and their implementing regulations, public procurement regulations and sector-specific regulations.

These impose additional requirements on individual public service companies. In addition, there are, of course, many other regulatory and non-normative regulators, often resulting from international obligations that determine the framework for the operation of state-owned enterprises (in almost all public service sectors). Consequently, the decisions of the shareholders, as well as the acts of the company's internal regulators and senior officials, are also relevant.

In fact, the fulfilment of all the rules and requirements that determine the purpose of the operation of state-owned enterprises is one of the issues—the deficiencies of which can give rise to the questionable validity of the given company. On the other hand, while the rules on the criminal liability of companies, as legal persons, and of executives and bodies have evolved significantly, prevention also plays a key role in the area of regularity [37]. The issue of adequacy and the development of the related control function at the legislative level in Hungarian law have mainly appeared in the field of credit institutions and insurance law for several years. CCXXXVII of 2013 on Credit Institutions and Financial Enterprises (CIFE): in January 2018, an amendment to the law introduced the obligation for CIFE-compliant entities to set up a legal compliance department. Similar rules appear in Act LXXXVIII of 2014 on Insurance Activities and in Act CXXXVIII of 2007 on Investment, Corporate and Commodity Services and on the Rules of Activities of Investment Firms and Commodity Providers (AIF). Useful guidance can be found in Recommendation 27/2018 (XII.10.) of the National Bank of Hungary (NBH) on the Establishment and Operation of Internal Security Lines, on the Management and Control Functions of Financial Organizations, and on the Establishment and Operation of Internal Security Lines. 5/2016. (VI. 6.) of the Hungarian National Bank and the Codex of Best Practice of the Hungarian Banking Association.

According to the aforementioned legislation, compliance control basically aims at examining the fulfilment of obligations and expectations appearing in legal and non-legal regulators, the actual operation purpose, values and principles of the organization and, if necessary, intervening in the processes in order to eliminate the deficiencies. This requires defensive lines and an internal procedure. They are designed to prevent negative effects on the body [17]. However, the full content of these cannot be covered by normative rules, and therefore, in the case of state-owned enterprises, ownership, control and the company's chief executive officer have a key role to play. This is also to be emphasized because the specific methods and rules of risk management applicable in a given organization cannot be determined by the legal regulators [38] and must always be adapted to the specific eligibility criteria of the organization concerned. In this process, the commitment of the leader of the organization to ensuring compliance is of paramount importance.

Company law, and in particular the legislation on state-owned enterprises, does not contain any guidance on compliance or sustainable corporate governance. Hungarian law is currently approaching these issues from the point of view of public financial regulation in the case of state-owned enterprises. Pursuant to Act CXCV of 2011 on Public Finances, the purpose of public finance controls is to ensure the regular, economical, efficient and effective management of public funds and national assets, and to ensure that reporting and reporting obligations are properly discharged. In accordance with our National Property Law, the property practitioner regularly monitors the management of the national property user, reports its findings to the national property user, and, whether its findings are within the competence of the State Audit Office (SAO). The SAO of Hungary annually audits the activities related to the exercise of ownership over state assets. In addition, the Hungarian National Asset Management Inc. regularly audits the management of the public assets by its contracted persons, entities or other

users, and reports its findings to the Hungarian National Asset Management Inc. Supervisory Board, the audited entity, the Secretary and the State Audit Office. The Public Finance Act identifies different levels of public finance control, such as the Treasury, the Government, the Government Audit Authority, the European Aid Audit Body and the Treasury in the external cases specified by the State Audit Office and the Public Finance Act on internal control functions in relation to those subject to it. With regard to internal control, Article 69 (1) of the Public Finance Act also states that the internal control system is a system of risk management and objective assurance designed to achieve the following objectives:

- to carry out the activities in the course of their operation and management in a regular, economic, efficient and effective manner;
- to comply with clearing obligations and protect resources from loss, damage and improper use.

The Public Finance Act also stipulates that the head of the budgetary body is responsible for the establishment, operation and development of the internal control system, taking into account the methodological guidelines published by the Minister for Public Finance. According to INTOSAI Internal Control Standards for the Public Sector [39], "Internal control is a dynamic, complex process that is constantly adapting to changes in the organization. Management and all levels of employees must be involved in this process in order to identify risks and provide reasonable assurance that the organization's mission and objectives will be met". In the Committee of Sponsoring Organizations of the Treadway Commission (COSO) model, internal control is a process that is influenced by the company's board of directors, management, and employees, and is designed to provide reasonable assurance of achieving organizational goals such as efficient and effective operation, internal and external financial reliability of reporting and compliance with applicable laws, regulations and internal regulations [40].

According to the framework defined by the Institute of Internal Auditors, "internal audit is an independent and objective assurance and advisory activity designed to improve the organization's operations and add value". Internal audit helps the organization achieve its objectives by implementing a systematic and disciplined concept to evaluate and improve the effectiveness of its risk management, organization management, and control processes [41,42]. But why did we refer to these rules? The relationship between public finances and state-owned enterprises is, in addition to the above-mentioned property law rules, essentially governed by Article 69A of the Public Finance Act, which applies to the internal control system of other entities classified in the general government sector. Other entities classified in the government sector are currently covered by a Ministry of Finance statement. Section 1 (12) of the Public Finance Act mentions the concept of another organization classified in the general government sector. This includes organizations which, under the Public Finance Act, are not part of the general government but which are covered by Council Regulation (EC) No 479/2009 of 25 May 2009 on the application of the Protocol on the excessive deficit procedure annexed to the Treaty on European Union they belong to the government sector. Other entities classified in the general government sector are subject to a number of obligations, as they are required to report to the Minister for Public Finances for the preparation of the Central Budget Act, and for the compilation required by the Central Budget Implementing Act. Government is required to provide regular data as specified in Government Decree 368/2011 (XII. 31.) on the implementation of the Public Finance Act. Pursuant to Article 9 of Act CXCIV of 2011 on Economic Stability of Hungary, a debt transaction may be entered into force with the prior consent of the Minister for Public Finances only, in accordance with Government Decree 353/2011 (XII. 30.) on the detailed rules for consenting to debt transactions.

The first part of this Communication contains other entities operating in the government sector at the time of publication, which are required to be fully enforced by law. Point (a) of the Communication deals with organizations classified in the Central Government subsector. Most of the nearly one hundred and fifty companies listed here are state-owned businesses. According to the preamble to Directive 2011/85/EU, "transparency is the key to ensuring that fiscal policy is based on realistic forecasts, not only through the use of official macroeconomic and budgetary forecasts for budgetary

planning but also the underlying methods, assumptions and relevant data. parameters should also be publicly available" [43]. Internal auditing "shall ensure that existing rules apply across the sub-sectors of general government. Independent auditing by public institutions such as the supreme audit institutions or private auditing organizations should encourage best international practice" [43].

The rules on internal control are laid down in subsection 47 of the Public Finance Act, while the detailed rules of the internal control system are laid down in Government Decree 370/2011 (XII. 31.) on the Internal Control and Internal Audit of Budgetary Organizations (Government Decree on Internal Control). It also applies to other entities. However, the Government Decree on Internal Audit does not address the issue of compliance but defines certain rules for the elements of the internal control system. Pursuant to Article 3 of the Internal Control Government Decree, the head of the budgetary organization shall be responsible within the internal control system for all levels of the organization:

- control environment,
- an integrated risk management system,
- control activities,
- information and communication system, and
- monitoring system design, operation and development.

In 2016, an additional element was added to the issue of organizational integrity based on Section 16 (g) of Government Decree 187/2016 (VII.13.) on the amendment of certain government decrees in connection with the development of the internal control system and the integrity management system. Accordingly, Article 6 (4) of the Government Decree on Internal Control requires the head of the budgetary organization to regulate the procedures for handling incidents of organizational integrity as well as the procedures for integrated risk management. In fact, therefore, one of the key issues of sustainable corporate governance in Hungarian state-owned enterprises has been clarified the issue of integrity close to the concept of private sector compliance [23,44]. Factors or groups of factors that influence corporate sustainability can therefore be distinguished based on the literature review carried out by the Initial Hazard Factors (IHFs), Hazardous Increasing Factors (HIFs), and Control Enhancement Factors (CEFs). These groups of factors can be divided into three target systems based on their explored goals. On the one hand, preferring compliance with legal requirements and owner's requirements during operation (IHFs). On the other hand, voluntary commitments, social and environmental compliance (HIFs), which are primarily related to the proper use of EU funds. Third, the effective operation of corporate compliance control (CEFs), which integrates the prevention of corruption, the maximization of social and environmental benefits and the long-term efficient use of resources. In the above interpretations, we can compare and sort the groups of factors based on their relation to sustainability.

## 4. Materials and Methods

Through the method of inductive analysis, we defined the aim of the study to reveal the relationships. Based on the integrity questionnaire compiled by the State Audit Office, three indices were calculated: the Initial Hazard Factors (IHFs), Hazardous Increasing Factors (HIFs), and Control Enhancement Factors (CEFs). Data from the State Audit Office of Hungary for 2016, 2017 and 2018 [43] for state-owned companies were used. The IHFs, HIFs and CEFs indices were analyzed separately for each year, and we followed the inductive analytical process to accurately track ownership and regulatory conditions, measure environmental and social benefits, and profitability and long-term sustainability. Based on the results of the surveys carried out each year, the input factors of the IHFs, HIFs and CEFs indices were classified. We have identified six impact indicators that have made it possible to assess each of the groups of factors in terms of sustainability. The Comparison of the Initial Hazard Factors (IHFs), the Hazardous Increasing Factors (HIFs), and the Control Enhancement Factors (CEFs) were performed on the basis of the selected six impact indicators using the Analytical Hierarchy Process (AHP). Based on this, we determined which group of factors dominated the implementation of

responsible corporate governance (CSR) with less risk or greater certainty. One of the basic tools for solving AHP decision problems is pairwise (pairwise) comparison, which is used to both weight the aspects and evaluate the alternatives according to each aspect. The pairwise comparison matrix is generally as follows, where $p_i$ ($i = 1, \ldots , n$) weights are any positive real numbers. If we write the pairwise comparison matrix for alternatives $A_1, A_2 \ldots A_n$, the matrix looks like this (Figure 1).

|        | $A_1$     | $A_2$     | $\cdots$ | $A_n$     |
|--------|-----------|-----------|----------|-----------|
| $A_1$  | $p_1/p_1$ | $p_1/p_2$ | $\cdots$ | $p_1/p_n$ |
| $A_2$  | $p_2/p_1$ | $p_2/p_2$ | $\cdots$ | $p_2/p_n$ |
| $\vdots$ | $\vdots$ | $\vdots$ | $\vdots$ | $\vdots$ |
| $A_n$  | $p_n/p_1$ | $p_n/p_2$ | $\cdots$ | $p_n/p_n$ |

**Figure 1.** Analytical Hierarchy Process (AHP) pairwise comparison matrix.

*In the matrix, $a_{ij} = p_i/p_j$ shos how many times the alternative $A_i$ is better than the alternative $A_j$.*

AHP is a powerful tool for complex decision making, helping decision makers to prioritize and make the best decision. AHP uses a series of paired comparisons to reduce complex decisions. Then, by synthesizing the results, it helps to capture both subjective and objective aspects. AHP is also used to reduce distortions in the decision-making process and contains a useful technique to check the consistency of the decision. The first step in solving decision tasks is to structure the decision task, which consists of defining the goal, choosing alternatives and defining aspects:

- 1. Goal: CSR operation;
- 2. Alternatives: IHFs, HIFs and CEFs;
- 3. Aspects: a, b, c, d, e, f:

　　a-　Integrated monitoring of legal regulators (LeMon),
　　b-　Fulfillment of profitability, size-efficiency (ProSize),
　　c-　Fulfill ownership requirements (OwnReq),
　　d-　Application of international standards (IntSta),
　　e-　Performance monitoring of social benefits (SociBen), and
　　f-　Successful use of EU and state resources (EUsour).

Note: Acronyms for each aspect or indicator are in parentheses.

We used the "Super Decision Software" to solve the decision task—in this case, the AHP model consists of the following steps:

*4.1. Creating the Test Matrix, Determining the Weights of the Aspects—The Weighting of the Selected Indicators (a, b, c, d, e, f) Based on the Three Groups of Factors (IHFs, HIFs and CEFs) (Table 1)*

**Table 1.** Comparative matrix of individual indicators and alternatives.

| | Alternatives<br>Indicators | Initial Hazard Factors (IHFs)<br>$(p_1 \ldots p_6)$ | Hazardous Increasing Factors(HIFs)<br>$(p_1 \ldots p_6)$ | Control Enhancement Factors (CEFs)<br>$(p_1 \ldots p_6)$ |
|---|---|---|---|---|
| a. | Integrated monitoring of legal regulators $(a_1 \ldots a_3)$ | $a_1/p_1$ | $a_2/p_1$ | $a_3/p_1$ |
| b. | Fulfillment of profitability, size-efficiency $(b_1 \ldots b_3)$ | $b_1/p_2$ | $b_2/p_2$ | $b_3/p_2$ |
| c. | Fulfill ownership requirements $(c_1 \ldots c_3)$ | $c_1/p_3$ | $c_2/p_3$ | $c_3/p_3$ |
| d. | Application of international standards $(d_1 \ldots d_3)$ | $d_1/p_4$ | $d_2/p_4$ | $d_3/p_4$ |
| e. | Performance monitoring of social benefits $(e_1 \ldots e_3)$ | $e_1/p_5$ | $e_2/p_5$ | $e_3/p_5$ |
| f. | Successful use of EU and state resources $(f_1 \ldots f_3)$ | $f_1/p_6$ | $f_2/p_6$ | $f_3/p_6$ |

Note: Indicators a, b, c, d, e, and f have been graded in alternatives $a_{1-3} \ldots f_{1-3}$ per row. The importance of the indicator examined for the selected alternative is indicated by the values $p_1 \ldots p_6$. The "p" value represents the highest value available for a given row for the goodness of the indicator, with possible values of 1, 3, 5, 7, and 9.

*4.2. The Process of Evaluating Alternatives According to the Criteria Given*

During decision making, the decision maker provides paired comparison matrices to determine the weights of the decision task aspect and evaluate alternatives (in this case: IHFs, HIFs and CEFs) for each leaf criterion or indicator (a, b, c, d, e, and f) (Figure 2). The importance of the indicators is divided into five grades, specifically "very unimportant", "less important", "important", "more important" and "very important", which are given a score of 1, 3, 5, 7, and 9, respectively.

Note: Initial Hazard Factors (IHFs) alternative rating chart, which also clearly shows that meeting owner needs plays a key role in this alternative.

*4.3. Summary of Weighting and Ratings*

By aggregating each level (1, 2, and 3), we obtain the values for the decision alternatives from which they can be ordered:

- 1CSR GOAL
- 2Clusters
- 3Alternatives

Note: The structure of the analysis and the design of the process can be seen in the Figure 3.

The results and the order of the analysis are used primarily to reduce distortions in the decision-making process. However, AHP also uses a useful technique to check the consistency of decisions. (Disadvantages of AHP: The disadvantage of pairwise comparisons is that they can only be applied below certain size limits for the objects to be compared, and only give rankings (relative

values) to alternatives. An advantage, however, is that they can be used very well in the evaluation of subjective aspects.)

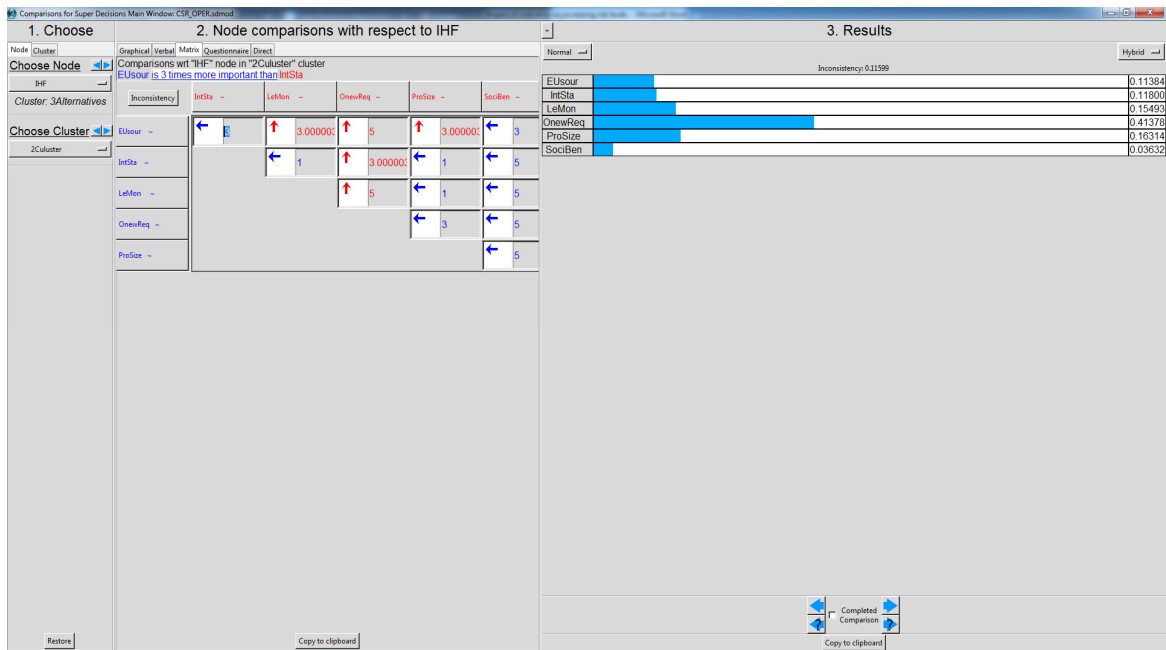

**Figure 2.** The process of the qualification of indicators.

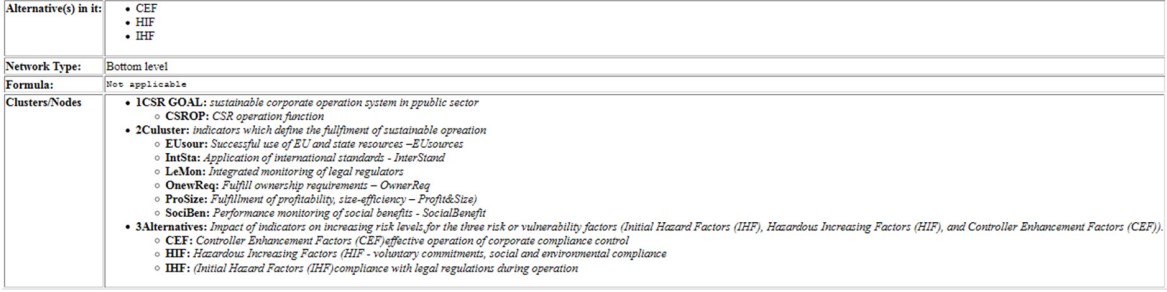

**Figure 3.** Structure of AHP analysis.

## 5. Results and Discussions

The practice of Hungarian state-owned enterprises is very heterogeneous. Due to their regulatory systems, a state-owned business entity must comply with numerous requirements in many areas of law. Because they are public property, they have to serve the public and spend money on the state, or "public", and they are increasingly challenged to monitor certain issues of compliance [45]. Based on this inductive analysis, we have identified those indicators that represent the assessment criteria that can be interpreted jointly for the Initial Hazard Factors (IHFs), Hazardous Increasing Factors (HIFs), and Control Enhancement Factors (CEFs) as alternatives. On the basis of the revealed indicators, we determined the relation of the individual indicators to each other and the order of decreasing the sustainability deficit of the groups of factors treated as alternatives.

### 5.1. Reasons for Inducing Sustainability Deficit and Their Inductive Analysis

The State Audit Office of Hungary has recently conducted a number of audits focusing on state-owned companies. The study, noted by the President of the State Audit Office and his co-authors, emphasizes that the deficiencies revealed by the audits are basically the lack of clear performance requirements, the lack of business plans and their contents, and shortcomings in the exercise of ownership control and supervisory rights, the lack of active management beyond the legal obligations.

In conclusion, the authors emphasize that "although the governance of state-owned enterprises is renewed, the growing economic environment provides more financial resources and opportunities if the governance and operation of public-owned enterprises are not regularly audited and audit findings have no consequences" [46]. Regarding the audit of public sector organizations, the implementation of the COSO model can be observed in Hungary from 2009—on the basis of which, a new system of rules of the internal control system of budgetary institutions has been elaborated. The subject matter and methods of auditing have become increasingly sophisticated in recent times and, along with financial auditing in the strict sense, performance-based and compliance audits have appeared. At the same time, this implies that besides the requirement to comply with and enforce normative requirements, there is a need to enforce and control certain (fundamental) principles, organizational goals, values or the intention of the legislator [47].

These normative and other criteria must necessarily appear in the documents of state-owned corporate regulators, as well as in strategic business planning and reporting. In the analysis of the State Audit Office, published in the summer of 2018, the operation of state-owned enterprises, the background documents and the findings of the audit reports of sixty-two state-owned enterprises published between 2015 and 2017 were processed. From the analyses, it is clear that there is a clear need to establish control mechanisms for the continuous measurement of the quality of the public tasks performed in order to strengthen ownership. According to the analysis, this is because the controls revealed a number of errors which indicate weaknesses in the control systems and, in the long run, generate sustainability deficits. Thus, for example, supervisory boards played a key role in auditing companies if the audit did not uncover many of the accounting irregularities highlighted by the Court's investigations. According to the analysis, only twenty percent of the companies applied ownership control measures in the field of asset management. It can also be pointed out from the SAO audit that the lack of profit and profitability represents the greatest sustainability risk for state-owned enterprises.

Risk management and risk assessment is an increasingly widespread issue for public sector organizations. Its conceptual definition has been examined by many authors, and although its sub-elements show differences in the views of individual disciplines, virtually every definition shows that risk is the possibility of a negative effect, which may affect the goals, the operation, the judgment, the trust in the organization in a negative way. Risks can be very diverse and although the negative impact we have mentioned can be expressed in some form of financial loss, it is not limited to financial value since it is difficult to measure and convert a trust deficit in an organization into a monetary measure, especially (e.g., waste management companies) where there is no other available service provider on the market [48]. The State Audit Office also examines integrity issues from year to year in relation to public sector institutions. Based on these, in 2014, it can be emphasized that corruption risks are mostly related to the preparation and conduct of public procurements, the use of EU funds, but the legal environment, the organizational structure and the changes in competences were also identified as risk factors.

In 2017, the state of integrity of public services was examined and it has been established that the provision of a fee-based service, the opportunity to exercise equity, excessive demand for the service carries a significant integrity risk and the complaint and was not appropriate everywhere. At the same time, there have been many advances compared to previous years, such as the issue of conflicts of interest, which are well regulated by most institutions, but have also highlighted a significant lack of integrity in the public sector, particularly in organizations where demand is consistently higher than supply. It is the closest to the issue of integrity used in public administration with regard to the purpose of "compliance'—both of which are intended to increase the resilience of the organization and to ensure its integrity [47–50]. The State Audit Office defines integrity as the totality of attributes, abilities, attitudes, and behaviors that are intended to serve the public interest and to ensure that public administration functions properly, effectively, and efficiently [24].

With regard to the integrity of state-owned enterprises, we examined the Hungarian State Audit Office's analysis of state-owned company integrity issues for 2016, 2017 and 2018 (State Audit Office,

2019) to identify areas of greatest risk report on the proper, sustainable operation of state-owned enterprises [49–52]. Based on the responses to the integrity questionnaire compiled by the State Audit Office, three indices (expressed as a percentage) of the integrity, corruption involvement of business entities, and the level of integrity controls performed were calculated (Table 2)—the Initial Hazard Factors (IHFs), Hazardous Increasing Factors (HIFs), and Control Enhancement Factors (CEFs) Index [49,53,54]. The comparative assessment for the years 2016, 2017, 2018 is presented in Sections 2–6. It is clear from Table 2 that HIF and CEF values also declined with the decrease in IHF values, which clearly means a decrease in the overall degree of integrity and cannot be called a positive trend.

**Table 2.** The IHFs, HIFs and CEFs indices reflect the integrity vulnerability of participantsin the 2016–2018 integrity survey and the level of control.

| Index | The Value of the Index in 2016 (%) | The Value of the Index in 2017 (%) | The Value of the Index in 2018 (%) |
|---|---|---|---|
| Initial Hazard Factors (IHFs) | 48.4 | 41.4 | 41.6 |
| Hazardous Increasing Factors (HIFs) | 35.2 | 25.0 | 27.8 |
| Control Enhancement Factors (CEFs) | 60.0 | 49.8 | 46.6 |

Based on Table 3 detailing the Initial Hazard Factors (IHFs) index, it is clear that the activities that represented a high risk in 2016 did not develop further among the companies' activities in 2017 and 2018 either.

**Table 3.** The Initial Hazard Factors (IHFs) index 2016–2018.

| Factors Qualifying for IHFs in 2016 | Factors Qualifying for IHFs in 2017 | Factors Qualifying for IHFs in 2018 |
|---|---|---|
| **Carrying out public tasks and pursuing other activities.** | Carrying out public and service tasks and carrying out other activities. The blending of the objectives and conditions of the Community mission and the operation of a competitive market has contributed to an increased vulnerability to integrity. | The mixed activity also involved other initial hazards. |
| **It affects the operation of business associations and the initial hazard to corruption**<br>**- the contents of the memorandum and articles of association, the management agreement, the operation contract, the public service delegation contract, and the technical and financial requirements relating to the performance of the tasks, and**<br>**- related ownership reporting. If these required clear and stringent requirements, the initial risk of corruption was less.** | More complicated and more complex legal environment. | The initial hazard of public service and high-balance sheet companies was almost double that of smaller, non-public service companies. |
| **As regards public service charges, the majority of the state-owned public service companies, which are entrusted with public service tasks, lack licensing or equity.** | | The operation of the public service and the size of the plant enhance the influence of each other. |

When examining the Hazardous Increasing Factors (HIFs), it is clear that the use and management of EU funds represent the greatest risk. In this case, the threat of corruption and the maintenance of programs, as well as the dominance and retention of environmental and social benefits, are high priority areas (Table 4). In 2018, the lack of competence related to the performance of professional tasks further increased the risks.

**Table 4.** The Hazardous Increasing Factors (HIFs) index in 2016–2018.

| Factors Qualifying for HIFs in 2016 | Factors Qualifying for HIFs in 2017 | Factors Qualifying for HIFs in 2018 |
| --- | --- | --- |
| **He has been a recipient of EU funding for the past three years, having been involved as a contracting authority, tenderer and in both ways in public procurement. Maintenance obligation, environmental compliance.** | He has received EU funding for the past three years and is now subject to public procurement obligations. Recruitment of an external expert/consultant for the activity, such as obtaining a grant or conducting a public procurement. Environmental compliance: the use of an environmental monitoring system is required to fulfill the application criteria. | Larger companies' overdue debts pose a greater risk. It was identified as a significant difference factor from a member company of the group: using a service or providing a service. Supporting other organizations or companies and employing or subcontracting external environmental consultants or experts. |
| **There was a significantly higher proportion of public service, public-service and other companies increasing the risk of corruption.** | The risk of companies using a restricted procedure in the field of public procurement was greater than that of firms not in the scope of public procurement. The continuous additional cost of defects in construction is a significant cost during maintenance. | Lack of approval by the supervisory board of the internal control plan was more often a risk-increasing factor for smaller companies. |
| **Increasing the risk of corruption when a company is involved in a public procurement procedure is because it uses a significant amount of public money.** | | The vulnerability of public service companies has already exceeded that of non-public service providers with a subsidy amount of EUR 3 million. Public service companies have a higher HIFs index, mainly because (in the absence of competence) they subcontract their core tasks more frequently, have recourse to a restricted procedure in their procurement and receive more state aid than non-public service companies. The use of the restricted procedure was accompanied by a higher incidence of other public procurement risk factors, such as several public tenders awarded to the same tenderer and the fact that the company was the successful tenderer with which the company was previously contracted. |

The risk reduction statement is illustrated by the evolution of Risk Mitigation Control Factors (RMCFs) (Figure 4). Analyses have shown that there is a positive relationship between the proportion of state ownership and the degree of the control of risk mitigating factors. Thus, the lower the proportion of state ownership, the better the sustainability aspects will be in the operation of the company. Here, income production is the most important aspect.

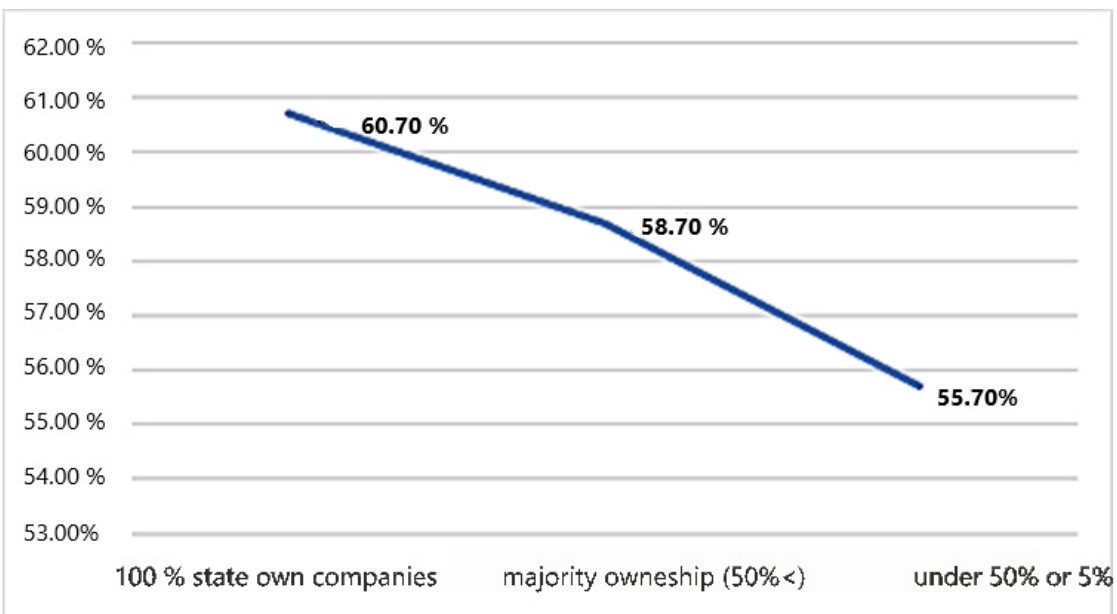

**Figure 4.** Average of Risk Mitigation Control Factor (RMCF) index by state ownership ratio.

Control enhancement factors were examined by the CEFs index. There is a distinction between so-called hard controls required by law and soft controls in non-normative regulators. There is a separate methodology for checking these, depending on the category. The methodology of control, although it is extremely important to improve it, is not addressed in this study. Control factors were examined by the CEFs index. The CEFs index continued to decline at 60% in 2016, reaching 49.8% in 2017 and only 46.6% in 2018. This change means that the number of factors under control has been steadily decreasing for the investigated companies. The level of development of both the factors under strong control and the factors in the soft control group decreased. In the case of large companies, the relationship between the management and the supervisory board has diminished, in the case of smaller companies, soft or social control has become almost insignificant (Table 5). For companies that did not receive EU support, the CEFs index was significantly higher.

Through the integration of integrity controls, we examine the level of corporate capabilities, attitudes, and behaviors that are intended to serve the public interest in a sustainable manner, to ensure that public administration functions properly, efficiently, and effectively. The intensity of the build-up of integrity controls responds to the degree to which companies have controls in each of the priority areas. Based on the characteristics collected in Table 6, it can be concluded that the proportion of named areas has steadily increased in recent years. However, this did not imply an increase in the introduction of integrity control systems. It can be pointed out from the analyses that the areas of the appearance of integrity were constantly expanding into different areas of social responsibility (introducing environmental responsibility and social utility measurement).

On the basis of our investigations, it can be emphasized that there was a risk to the management of the companies, that the legal obligations did not oblige the companies, the assigned owners' practitioners to pre-fix in contract the expected level of performance of public tasks. Typical errors in the audited companies include the fact that the contracts for the performance of the tasks were not brought into line with the changes in the legislation, and only 20% of the companies were subject to on-the-spot checks of the management. Based on the analysis of the State Audit Office, it can be concluded that most of the companies have established the controls necessary for the management of companies. However, there were lags in certain areas of external and internal audits and organizational culture. The results of the 2017 survey show that the existence and professional operation of internal control is one of the controls that has a positive impact on the integrity of the company as a whole.

Almost all state-owned enterprises have had accounting irregularities identified by the State Audit Office during their audits but have not been detected by independent auditors before.

**Table 5.** Control Enhancement Factors (CEFs) index in 2016–2018.

| Factors Qualifying for CEFs in 2016 | Factors Qualifying for CEFs in 2017 | Factors Qualifying for CEFs in 2018 |
| --- | --- | --- |
| **The relationship between size and hazard has not been studied in detail**. | In terms of control structure, the largest differences were found between the total assets, the size of the balance sheet, the reporting activity and the internal control function. | The absolute value of the control structure was significantly influenced by the plant size. |
| | Higher risk was associated with higher levels of control. For assets with a balance sheet total of more than EUR 2 million, the level of controls is above average. | There was a significant difference in the intensity of the control factors for companies below and above the balance sheet total of EUR 2 million. Larger companies had better controls. |
| | If the management (director/chief executive/executive) of a company did not or regularly inform the supervisory board of the decisions taken by the management and their implementation, this was also related to the lower level of control in place. | The intensity of the soft controls is only moderate in the larger companies, but very low in the smaller companies. |
| | If a company did not have a risk-based internal control plan, it had a below average control system, which was also significantly lower than companies with an internal control plan based on risk analysis. | If a company received an EU subsidy, it increased the CEFs index by an average of 6.3 percentage points (index of EU subsidized companies: 51.1%, of non-subsidized companies: 44.8%). |

Factors that increase the risk of corruption have been above average due to the occurrence of EU subsidies, public procurement, organizational and regulatory changes. At the same time, the level of controls (68.5%) reducing the risk of corruption was already above average, according to a 2015 survey, which improved by 2017. Nearly 60 percent more companies participated in the 2018 integrity survey than in 2017. Based on the processing of the questionnaires, the State Audit Office found that the average exposure of public service companies was higher than that of market service companies. In these cases, the public service was often accompanied by circular feedbacks (e.g., asset management, performing a public authority task, using financial support) that increased the integrity risk. Higher sensitivity in public service companies is offset by a stronger than average control system. It is also important to underline, according to their research, that the increase in the size of companies is accompanied by an increase in their sensitivity. Belonging to a group of companies also slightly increased the vulnerability of companies, which could be compensated by the use of group-level controls. It is a particularly disadvantageous feature of the surveys that most of the investigated companies lack the feedback related to the performance of their activities, which primarily concerns the social impact, the achieved environmental performance and the level of activity.

Although we can find indicators of income-centric, socially efficient management among the Control Enhancement Factors, there is no corporate target system for their application. For non-profit-oriented companies, it is difficult to define target indicators that measure true corporate performance. New business models and operating principles appearing in the circular economy concept do not add another dimension to corporate control. The analyses carried out show that the preference for compliance with the law and the requirements of the owners, the compliance with the social and environmental requirements in operation and the conditions for the proper use of EU funds,

and the maximization of social and environmental benefits in the long term can also be implemented through a properly integrated control system.

**Table 6.** Areas of application of integrity controls in 2016–2018.

| Areas of Application of Integrity Controls 2016 | Areas of Application of Integrity Controls 2017 | Areas of Application of Integrity Controls 2018 |
|---|---|---|
| - Asset management, management of public funds;<br>- Corporate governance, supervision;<br>- Activities, provision of public services;<br>- Organizational structure;<br>- Purchases, public procurement;<br>- Legal environment;<br>- Internal regulation;<br>- Human resource management;<br>- Internal controls, risk management;<br>- Special anti-corruption systems and procedures. | - Responsible management (provision of ownership, legal environment, role of the supervisory board, reporting to the owner, decision-making power of the owner, corporate governance, establishment of the organizational structure, management information system);<br>- Public service tasks, external relations (public service tasks, public service provision, fee setting, support to and from outside organizations, risks of contractual partners, outsourcing, disclosure, publicity);<br>- Management (resource and asset management, EU support, partner contracts, group of companies, management efficiency);<br>- Compliance, audits (internal regulations, public procurement, tendering, auditing, external and internal audits);<br>- Organizational culture, ethical behavior (integrity culture in internal rules, private benefits, staff selection, conflict of interest, benchmarking, ethics, media appearance). | Beyond the areas named in the 2017 analysis:<br>- External relations: Participation in, or provision of, external aid, accounting for subsidies, measurement of customer satisfaction and social utility, disclosure of data of public interest.<br>- External and internal audits: auditing, audits by external bodies, quality of internal audits, utilization, environmental and financial risk analysis and risk management.<br>- Organizational culture: human resources management, conflict of interest management and employee selection, remuneration system, performance evaluation, ethical procedures. |

Based on the analysis, it has been established that in the course of the operation of the company, the Initial Hazard Factors (IHFs), Hazardous Increasing Factors (HIFs), and Control Enhancement Factors (CEFs) index can be examined by changing the following indicators:

- Integrated monitoring of legal regulators;
- Fulfilment of profitability, size-efficiency;
- Fulfilled ownership requirements;
- Application of international standards;
- Performance monitoring of social benefits;
- Successful use of EU and state resources.

The surveys also give us an idea of the extent to which the relationship between the indicators and the index systems are leading towards corporate sustainability. We used the Analytic Hierarchy Process (AHP) method to examine the relationship between the indicators and to determine the goodness of each index (IHFs, HIFs, CEFs).

*5.2. The Results of Analytic Hierarchy Process (AHP) Comparative Analysis*

The Analytic Hierarchy Process was used to determine relationships and the order of factors. AHP is a powerful tool for complex decision-making and helps decision-makers to prioritize and make the best decision. AHP uses a series of paired comparisons to reduce complex decisions. Then, by synthesizing the results, it helps to capture both subjective and objective aspects. In the AHP matrix we set up, the indices revealed by inductive analysis were analyzed according to their importance for each alternative (IHFs, HIFs, CEFs) (Table 7).

**Table 7.** AHP comparative matrix of the research.

| | Indicators | Alternatives | Initial Hazard Factors (IHFs) | Hazardous Increasing Factors (HIFs) | Control Enhancement Factors (CEFs) |
|---|---|---|---|---|---|
| a) | Integrated monitoring of legal regulators (LeMon) | | 5/7 (0.71) | 1/7 (0.14) | **7/7 (1.0)** |
| b) | Fulfillment of profitability, size-efficiency (ProSize) | | 5/7 (0.71) | 5/7 (0.71) | **7/7 (1.0)** |
| c) | Fulfill ownership requirements (OwnReq) | | **7/7 (1.0)** | 3/7 (0.42) | 5/7 (0.71) |
| d) | Application of international standards (IntSta) | | 5/7 (0.71) | **7/7 (1.0)** | **7/7 (1.0)** |
| e) | Performance monitoring of social benefits (SociBen) | | 3/9 (0.33) | **9/9 (1.0)** | 5/9 (0.55) |
| f) | Successful use of EU and state resources (EUsour) | | 3/7 (0.42) | 5/7 (0.71) | **7/7 (1.0)** |

Note: The evaluation of the pairwise comparison is shown in the table. Quantification of the specified value (between 0.0 and 1.0) is the number in parentheses "7/7 (1.0)".

Based on the comparative matrix data, we can see that for IHFs, indicator c), for HIFs, indicators d) and e), for CEFs, indicators a), b), and d) are the most weighted indicators in the analyzed list. The results of the matrix show that the measurement of social benefits and the environmental aspects related to the use of EU funds have the lowest weight in the case of IHFs. Initial Hazard Factors (IHFs) are focused on satisfying the interests of its owners, which in many cases overrides profitability and the importance of strictly observing accounting rules. This property is stronger the higher the proportion of state ownership in the ownership share of the enterprise. In the case of Hazardous Increasing Factors (HIFs), it should be emphasized on the basis of AHP studies that the application of international standards and the measurement of social and environmental benefits will be the focus of the study alternative. The application of the HIFs index does not focus on the interests of the owners or on strict compliance with the accounting regulations. However, it is interesting to note that sustainable income generating ability or corporate size efficiency is one of the most important considerations when measuring index performance.

In the case of the Control Enhancement Factors (CEFs) index, accounting rules and control factors that influence legitimate operations, profitability and size efficiency are all in focus due to the integrated approach. It should be noted that the CEFs index focuses on compliance with international standards and effective and regular use of EU funds. It is important to emphasize that the measurement of social benefits is not one of the most important indicators, but the system of criteria applied in the use of EU funds can basically fulfill this requirement if there is use of EU funds. The final result of the "Super Decision software" consolidated evaluation, which analyzed the optimal conditions for Corporate Social Responsibility (CSR) operation through a pair-wise comparison of indicators, the two most important areas affecting sustainable operation are "Integrated monitoring of legal regulators" and "Performance monitoring of social benefits" (Figure 5).

The results show that state-owned enterprises can operate in the longest, most sustainable way through the rule-based corporate management system, controlling and maximizing social benefits (presented by Figure 5 and Appendix A).

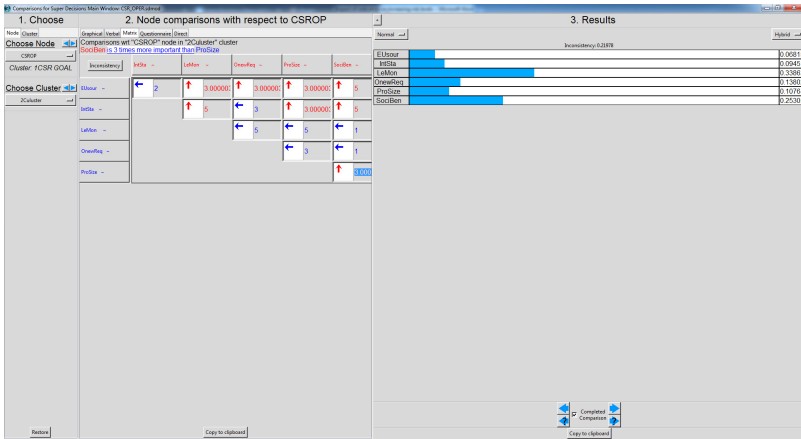

**Figure 5.** The CSR goal, the importance of the areas that determine the operation.

From Figure 6, it can be clearly seen that, among the alternatives, the study determined the order of (1) CEFs - (2) HIFs - (3) IHFs, which indicates that the Control Enhancement Factors (CEFs) index is closest to the best measurement system to determine the principles of sustainable business operation. With a view to enhancing integrity, CEFs is not only a suitable measurement system for the fulfilment of public functions and ownership needs, but also for income generation and environmental/social needs that favor market operation. Investigations have also shown that the key to the operation of state-owned enterprises is the management and professional use of EU funds. It is a recurring problem, not only for state-owned enterprises but also for other public sector organizations, that EU support and the conduct of public procurement procedures are accompanied by an increase in integrity threats, even if the organizations try to counteract this risk by reinforcing the control system. Our research also highlighted the interesting point that the average level of control over companies with 100% public ownership is achieved jointly by several owners. In other words, the notion that a single owner does not need to have full control over corporate governance is erroneous. The rules for this should be elaborated not by the company but by the owners. Company size means optimum control design depending on the functionality of the enterprise units. Finally, companies that have the lowest level of control do not evaluate the achievement of goals set for the company's efficiency and effectiveness, and do not have a reporting system to manage social, environmental issues, and measure and monitor corporate CSR performance.

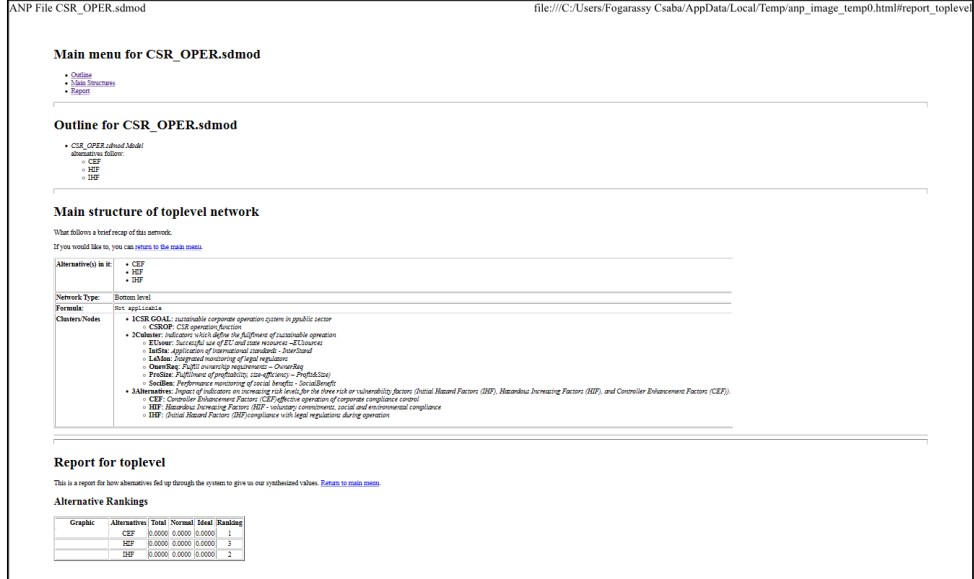

**Figure 6.** The structure of the analysis and the order of the alternatives (IHFs, HIFs, CEFs).

## 6. Conclusions

The issue of corporate sustainability is essential for current business and is actually a tool for future business associations. Corporate sustainability has many segments. In our study, one of these was chosen as one of the segments of economic sustainability—the issue of corporate compliance. Based on our research, we have found that compliance with public law compliance in the Hungarian legal system can be identified with the integrity of state-owned companies. Although a well-known term, state-owned business associations do not have any guiding legislation in terms of terms, but they also have to apply some rules on the integrity of public administrations to companies. Our analysis is based on the Hungarian State Audit Office audits which revealed a number of errors that are systemic deficiencies and generate sustainability deficits in the long run. Analyzing the results of the State Audit Office audits, we conclude that the lack of profit and profitability is the biggest sustainability risk for state-owned enterprises. Thus, after the end of the external funds, there are no financial instruments, adequate reserves for the operation of the established practice or asset, so these services and equipment previously operated from the sources of support cannot be maintained. According to our studies, it can be clearly stated that in the current system of conditions, the size of the company clearly increases the risk of corporate sustainability (long-term stable operation), which is also endangered by the volume of external resources, i.e., the larger the amount of external support resources entered into the enterprise operation system, the more they are more vulnerable to sustaining sustainable operations. The issue is somewhat complicated by the fact that companies need to meet a much more segmented and more complex expectations system than a budget entity, so we believe it is essential to regulate the control mechanisms of this very important segment of corporate sustainability at the legislative level. In our study, we examined the practice and integrity analyses of the State Audit Office for the control of state-owned enterprises. As far as the sustainability aspects of corporate integrity are concerned, it was already highlighted in 2014 that corruption risks are mainly related to the preparation, conduct of public procurement and the use of EU funds (although the situation improved significantly until 2018, mainly due to the correct identification of conflicts of interest). We are well aware that the reduction of risk factors can be a systematic check/self-check (embedding into a closed electronic system) or a combination of EU funds (subsidy + bank credit). If the leakage of financial assets that are not related to the operation of certain long-term corporate functions is high, maintaining corporate sustainability is risky. On the basis of the above, it can be clearly concluded that the control mechanisms closely interacting with each other should be directed to a common platform. Minimally, a framework law should set out organizational integrity and compliance requirements, control procedures and sanction systems for corporate sustainability. Based on the literature analysis and our research findings, we have identified a number of factors (including the size of the business and intensive use of EU funding resources) that may result in the loss of corporate sustainability. A number of risk-increasing factors have been identified which are controversial in social terms (including the greater the use of EU money, the greater the level of corruption). These threats may be affected by the effective internal control environment we investigate and recommend for deployment. Our analysis shows that most of the Hungarian phenomena are characteristic of the former socialist countries of Central Europe as well. In many cases, these sustainability risks also entail economic and political risks. The interpretation of sustainability in countries with economies in transition, where market economy and state involvement are random and protectionist, is very different from the most developed countries in the European Union. In these first-generation businesses, which have evolved in a socialist market economy, generational rules for successive generations do not yet exist.

Limitations: We did not use endogenous or exogenous growth theory models, nor did we consider the impact of sustainability factors on resource efficiency. The development of human resources or the increase in R&D were not among the aspects of increasing the compliance factor.

**Author Contributions:** Conceptualization, A.B; Data curation, C.F.; Formal analysis, C.F.; Investigation, A.B.; Methodology, C.F.; Supervision, C.F.; Writing—original draft, A.B.; Writing—review and editing, A.B.

**Funding:** This research received no external funding.

**Acknowledgments:** The preparation of the manuscript and our final article was supported by the Institute of Public Administration and Law, National University of Public Service, Climate Change Research Centre and Doctoral School of Management and Business Administration at Szent Istvan University.

**Conflicts of Interest:** The authors declare no conflict of interest.

## Appendix A

*The "Super Decision Program" evaluation matrix for CSR Goal, IHF, HIF and CEF Clusters.*

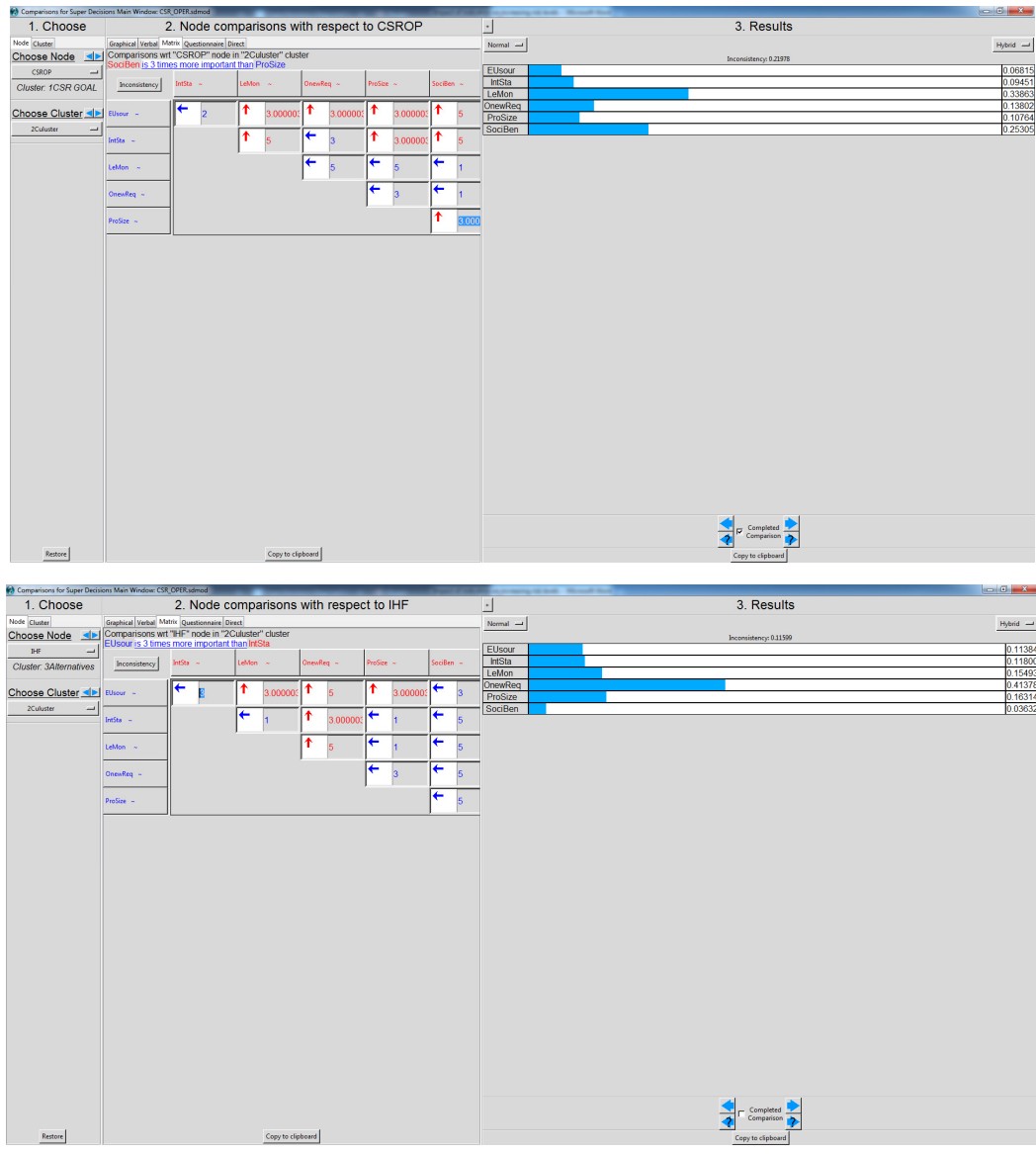

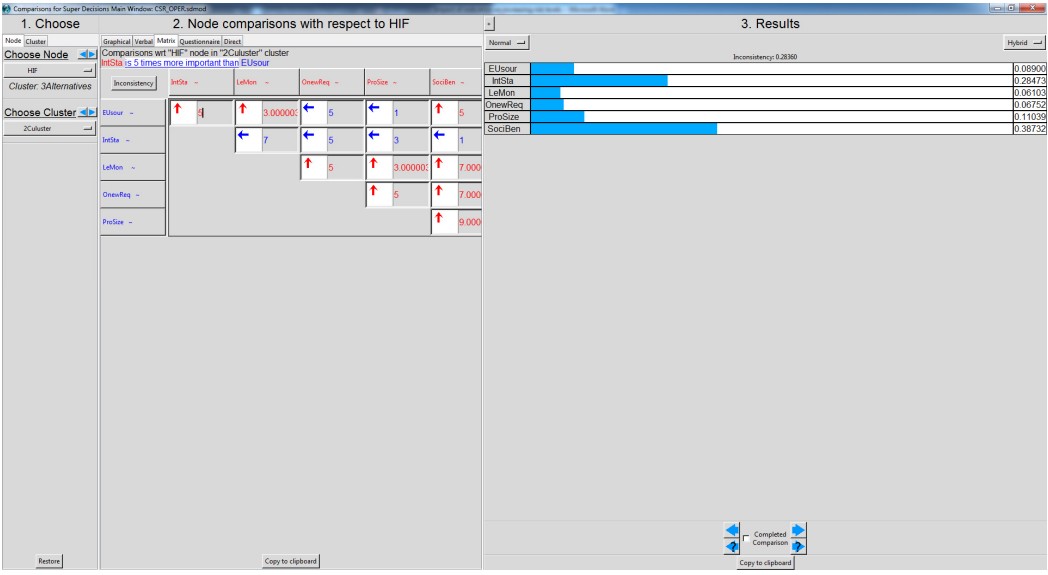

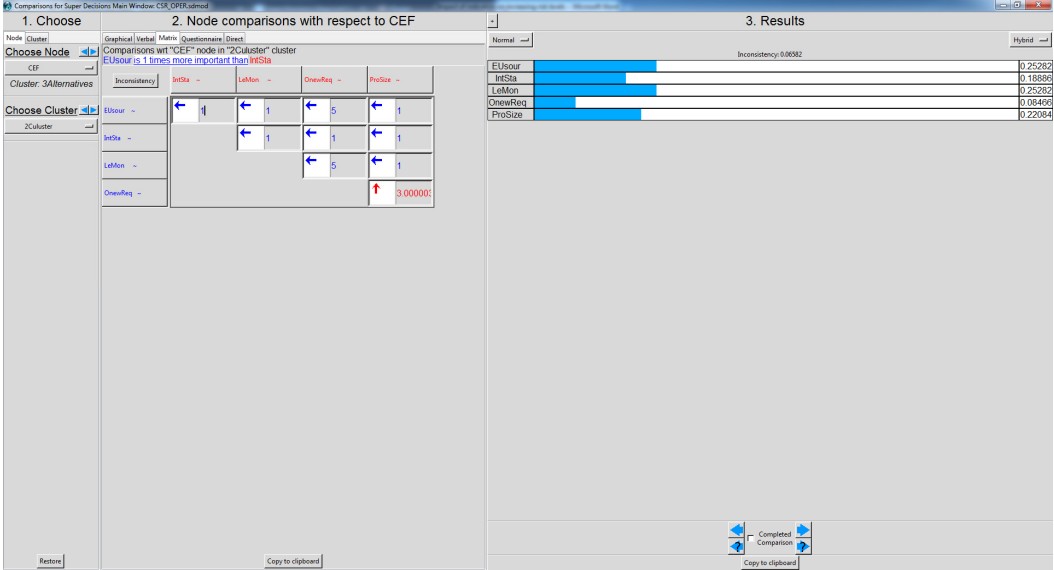

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
