# Peer review of "Relationship between Corporate Sustainability and Compliance with State-Owned Enterprises in Central-Europe: A Case Study from Hungary"

_sustainability, doi:10.3390/su11205653_

Round 1

Reviewer 1 Report

Thank you for giving me the opportunity to read your paper.

This study deals with the control of Hungarian state-owned business associations in order to find out whether there is any correlation between corporate sustainability and compliance.

The abstract is informative and, in general, it reflects the content of the paper. However, the relevance of the problem, in our opinion, should be reflected more fully. The tools for resolving the identified problems are not clearly indicated in the abstract, the novelty of the results is not determined.

In the “Introduction” section, the research object indicated in the title of the article is defined only indirectly: the word “Hungary” does not appear in the Introduction.

The analysis of significant and modern works on the conformity of the purpose of the study is made. At the same time, the absence of clearly defined tasks in the Introduction did not fully determine the detailed aspects of assessing the degree of this correspondence.

The choice of analysis methods is well founded. This allowed taking into account the significant factors (IHF, HIE, CEF), which were used in indexed form for testing. To rank and evaluate risk avoidance options, the useful AHP method was used.

It should be noted that the authors use the necessary empirical data to justify the use of the proposed analysis methods. The use of matrices that take into account the main risk factors made it possible to ensure sufficiently effective control of corporate sustainability regulation processes in the concept of the consent.

Of practical value are detailed descriptions of the characteristics of factors (tables 3,4,5). For Hungarian corporations, the recommended control integration zones and the necessary management actions for the implementation of the corporate sustainability process have been identified.

The results of the study are described by the authors in detail and specifically.

The bibliography contains relevant sources necessary to determine the degree of solution of the problem under study.

Author Response

Dear Reviewer,

Thank you very much for your appreciation and kind words. The shortcomings highlighted in the introductory section, which are highlighted in red in the text, have been corrected.

The Authors

Reviewer 2 Report

The manuscript is interesting, scientifically structured, and of important socio-economic and managerial applicability to the wider Central Europe context. Besides, insightful conclusions upon “corporate sustainability” are adding value to the analysis conducted. In this respect the manuscript sustains novel features and it can be accepted for publication at the “Sustainability” journal after the consideration of the following review comments.

1) At the “Introduction” section it is anticipated authors to provide a more pluralistic viewpoint of socio-environmental impacts of corporate sustainability at both enterprise level and organizational context (including the circular economy conceptualization and the quality of life improvement) in Europe and, to this end, the indicative list of relevant published papers below can be considered and cited. These extra citations can be cross-cited in the text extracts where only one citation per statement is positioned (this comment applies, generally, to the whole text content).

2) The methodology applied is robust and it contains a high level of novelty, being structured under the following framework:

a) The analysis of the IHF, HIF and CEF indices at the period 2016-2018, b) The deployment of the inductive analytical process to accurately track ownership and regulatory conditions, measure environmental and social benefits, and profitability and long-term sustainability. c) the selected six impact indicators using the Analytical Hierarchy Process (AHP). d) the determination of group of factors dominated the implementation of responsible corporate governance (CSR) with less risk or greater certainty.

However, at the methodology applied and based on the narrative deployment, the following critical aspects should be precisely addressed at the revised version:

a) The formulation of Tables and Figures has been bounded on indices and factors’ notation, while it is not straightforward which services/products/physical ontologies are derived from the “state-owned companies” mentioned? These companies sustain, probably, distinct features to each other, thus, diversified corporate sustainability “footprint” is anticipated. Therefore, a list of such typical ontologies/products served (no brand names) can be given in an extra Table. b) Upon analysis conducted, it seems that four significant aspects of corporate sustainability are missing, or undermined: firstly, the technological advancements and innovations’ spur, secondly, the extroversion and internationalization perspectives offered upon a liberalized/free/open international marketplace – that is implying the bilateral trading agreements/contracts, thirdly, the public responsibility and the economic feasibility of these state-owned companies economic to pair the Circular Economy Valuation (CEV) with Ecosystems Services Valuation (ESV), secondly, the analysis has been developed upon a narrow timeline of three subsequent years, whereas the social-, economic-, and environmental- contexts of analysis upon organizational compliance, they are necessitating institutional reforms/reflections that are anticipated in a medium- or long- term timeframe. Therefore, outcomes’ forecasting and “argumentative extrapolation” to 5-10 years future time it is highly recommended. I would like to stress out that each one of the aforementioned four aspects is unevenly affecting the “corporate sustainability” (being time-sensitive and spatial-/national-specific aspects). Therefore, there is no need author to deploy an extensive analysis, but to include them at the revised manuscript in a plenary-, cross-cited-, and descriptive- manner.

I would like to explicitly note that there is no need author to change or modify the existing narrative flow but, based on the aforementioned comment 2, to reinforce the creative discussion of their research outcomes. In this respect, up to one extra and cross-cited text page it is adequate. To this end, the following list of papers can be further considered.

Scopus

EXPORT DATE:11 Sep 2019

Taliento, M., Favino, C., Netti, A.

57190750126;55887073800;57208002242;

Impact of environmental, social, and governance information on economic performance: Evidence of a corporate 'sustainability advantage' from Europe

(2019) Sustainability (Switzerland), 11 (6), art. no. 1738, . Cited 3 times.

https://www.scopus.com/inward/record.uri?eid=2-s2.0-85063497821&doi=10.3390%2fsu11061738&partnerID=40&md5=1a31283afe70280b359a135d9bd983da

DOI: 10.3390/su11061738

AFFILIATIONS: Department of Economics, University of Foggia, Foggia, 71121, Italy

ABSTRACT: Both UN Agenda 2030 and the Directive n. 2014/95/EU have recently promoted a marked improvement in sustainability disclosure, especially for larger companies or groups. Starting from this premise, we carried out an original study on the financial materiality of the E-S-G (environmental, social and governance) information of primary companies listed on major European indices in Belgium, France, Germany, Italy and Spain (BEL, CAC, DAX, FTSE-MIB, IBEX). Within the Stakeholder Theory and the Corporate Social Responsibility (CSR)-Corporate Social Perfomance (CSP) framework, our empirical analysis examined the impact of non-financial results (assessed through sustainability indicators) on economic (financial and market) performance in the timespan 2014-2017. We propose a different approach from previous studies, based on a PLS (Partial least squares)/SEM (Structural equation modeling) methodology together with the unprecedented consideration of "ESG" measures (Environmental, Social and Governance), either absolute (scores) or relative (extra-performance over industry sector). We find that, despite the absolute level of the individual ESG scores not being impactful, the "distance" from the industry average-normal figures (excess or abnormal ESG performance) is positively relevant, collaterally revisiting the notion of competitive advantage in sustainability terms. Corporate size is shown to be a significant background factor (as slack resources proxy). Social, environmental and governance responsibility (to all stakeholders) appear to be important as a competitive factor of the modern firm. © 2019 by the authors.

AUTHOR KEYWORDS: Corporate governance;  Corporate size;  Environment;  ESG metrics;  Financial performance;  Social;  Stakeholder Theory;  Structural Equations Model;  Sustainability disclosure

DOCUMENT TYPE: Article

PUBLICATION STAGE: Final

ACCESS TYPE: Open Access

SOURCE: Scopus

Aravossis, K.G., Kapsalis, V.C., Kyriakopoulos, G.L., Xouleis, T.G.

6603320854;56124408200;6603382498;57210737455;

Development of a holistic assessment framework for industrial organizations

(2019) Sustainability (Switzerland), 11 (14), art. no. 3946, .

https://www.scopus.com/inward/record.uri?eid=2-s2.0-85071345908&doi=10.3390%2fsu11143946&partnerID=40&md5=3501b754c240eb4e508e2d93c2fbbee7

DOI: 10.3390/su11143946

AFFILIATIONS: Environmental Economics and Sustainability Unit, Sector of Industrial Management and Operations Research, School of Mechanical Engineering, National Technical University of Athens, Athens, GR 157 80, Greece;

GREENiT Environmental, Athens, GR 104 33, Greece

ABSTRACT: The evaluation and selection among the best production practices beyond the conventional linear models is, nowadays, concerned with those holistic approaches drawn toward environmental assessment in industry. Therefore, researchers need to develop an analysis that can evaluate the performance of industrial organization in the light of their environmental viewpoint. This study implemented a pilot co-integrated scheme based on an innovative in-house Holistic Assessment Performance Index for Environment (HAPI-E) industry tool while assimilating the principles of circular economy through the Eco-innovation Development and Implementation Tool (EDIT). For the latter, nine qualitative indicators were motivated and enriched the weighting criteria of the questionnaire. The decomposition of the complexity and preferences mapping was accompanied by a multi-criteria holistic hierarchical analysis methodology in order to synthesize a single index upon a need-driven scoring. This multi-criteria decision approach in industry can quantify the material and process flows, thus enhancing the existing knowledge of manipulating internal resources. The key-criteria were based on administrative, energy, water, emissions, and waste strategies. Subsequently, the HAPI-E industry tool was modeled on the food industry, being particularly focused on pasta-based industrial production. Then, the parameters of this tool were modeled, measured, and evaluated in terms of the environmental impact awareness. The magnitude of necessary improvements was unveiled, while future research orientations were discussed. The HAPI-E industry tool can be utilized as a precautionary methodology on sustainable assessment while incorporating multifaceted and quantification advantages. © 2019 by the authors.

AUTHOR KEYWORDS: Circular economy;  Environmental indicators;  Holistic assessment framework;  Proactiveness;  Sustainable development;  Sustainable production schemes

DOCUMENT TYPE: Article

PUBLICATION STAGE: Final

ACCESS TYPE: Open Access

SOURCE: Scopus

Kapsalis, V.C., Kyriakopoulos, G.L., Aravossis, K.G.

57208710234;6603382498;6603320854;

Investigation of ecosystem services and circular economy interactions under an inter-organizational framework

(2019) Energies, 12 (9), art. no. 1734, . Cited 1 time.

https://www.scopus.com/inward/record.uri?eid=2-s2.0-85066053518&doi=10.3390%2fen12091734&partnerID=40&md5=bf3293e829108db164ed6dfea9c31a6d

DOI: 10.3390/en12091734

AFFILIATIONS: School of Mechanical Engineering, Sector of Industrial Management and Operations Research, National Technical University of Athens, 9 Heroon Polytechniou Street, Athens, 15780, Greece;

School of Electrical and Computer Engineering, Electric Power Division, Photometry Laboratory, National Technical University of Athens, 9 Heroon Polytechniou Street, Athens, 15780, Greece

ABSTRACT: Nowadays, the conceptualization of circular economy is an attractive managerial tool among governments and businesses throughout the word, while ecosystem services are a contentious issue due to the particular needs of humans' well-being. At this review the interactions between the principles of ecosystem services and the circular economy were investigated in the light of inter-organizational systems. This evaluation was based on more and more complex processes, while the integration of the growing circular economy concept within the shrinking parent ecosystem unveiled challenges and constraints for products' end of life and quality. It was argued that: (a) The existence of social and people-related barriers can be considered under three groups, namely, the "sustainable provision and modeling schemes", "socio-cultural appreciation and payment schemes", and "regulatory and maintenance schemes", (b) The impacts of circular economy-ecosystem services toward an inter-organizational functional stream model associated with distinguished proactive and post treatment risk values (c) The functionality and the accountability of the technosphere are the two critical components to support the restorative and the regenerative perspectives of the biosphere. The aforementioned findings unveiled new emerging paths to be further investigated, offering a deeper appraisal of circular economy under the inter-organizational perception. © 2019 by the authors. Licensee MDPI, Basel, Switzerland.

AUTHOR KEYWORDS: Biosphere;  Circular economy;  Ecosystems services;  Energy flows;  Inter-organizational functional stream model;  Technosphere

DOCUMENT TYPE: Article

PUBLICATION STAGE: Final

ACCESS TYPE: Open Access

SOURCE: Scopus

Ntanos, S., Kyriakopoulos, G., Chalikias, M., Arabatzis, G., Skordoulis, M., Galatsidas, S., Drosos, D.

57076831500;6603382498;23092968200;8884728500;56251883100;6504287685;56394701400;

A social assessment of the usage of Renewable Energy Sources and its contribution to life quality: The case of an Attica Urban area in Greece

(2018) Sustainability (Switzerland), 10 (5), art. no. 1414, . Cited 14 times.

https://www.scopus.com/inward/record.uri?eid=2-s2.0-85046675238&doi=10.3390%2fsu10051414&partnerID=40&md5=a82ada3e2b02cdcf2d6859cbb11fc161

DOI: 10.3390/su10051414

AFFILIATIONS: Department of Forestry and Management of the Environment and Natural Resources, School of Agricultural and Forestry Sciences, Democritus University of Thrace, Orestiada, 68200, Greece;

School of Electrical and Computer Engineering, National Technical University of Athens, Zografou, 15780, Greece;

Department of Tourism Management, School of Business, Economics and Social Sciences, University ofWest Attica, Egaleo, 12244, Greece;

Department of Business Administration, School of Business, Economics and Social Sciences, University ofWest Attica, Egaleo, 12244, Greece

ABSTRACT: The aim of this paper is to analyze and evaluate the use of Renewable Energy Sources (RES) and their contribution to citizens' life quality. For this purpose, a survey was conducted using a sample of 400 residents in an urban area of the Attica region in Greece. The methods of Principal Components Analysis and Logit Regression were used on a dataset containing the respondents' views on various aspects of RES. Two statistical models were constructed for the identification of the main variables that are associated with the RES' usage and respondents' opinion on their contribution to life quality. The conclusions that can be drawn show that the respondents are adequately informed about some of the RES' types while most of them use at least one of the examined types of RES. The benefits that RES offer, were the most crucial variable in determining both respondents' perceptions on their usage and on their contribution to life quality. © 2018 by the author.

AUTHOR KEYWORDS: Life quality;  Logit regression;  Renewable energy sources;  RES public acceptance

DOCUMENT TYPE: Article

PUBLICATION STAGE: Final

ACCESS TYPE: Open Access

SOURCE: Scopus

Jones, P., Hillier, D., Comfort, D.

57209625905;55253136400;8064675100;

Materiality and external assurance in corporate sustainability reporting: An exploratory study of Europe’s leading commercial property companies

(2016) Journal of European Real Estate Research, 9 (2), pp. 147-170. Cited 2 times.

https://www.scopus.com/inward/record.uri?eid=2-s2.0-84983537444&doi=10.1108%2fJERER-07-2015-0027&partnerID=40&md5=326e490e23382e54d4be15730afc45e3

DOI: 10.1108/JERER-07-2015-0027

AFFILIATIONS: The Business School, University of Gloucestershire, Cheltenham, United Kingdom;

Centre for Police Sciences, University of South Wales, Pontypridd, United Kingdom

ABSTRACT: Purpose: The purposes of this paper are to provide a preliminary examination of the extent to which Europe’s leading commercial property companies are embracing the concept of materiality and commissioning independent external assurance as part of their sustainability reporting processes and to offer some wider reflections on materiality and external assurance in sustainability reporting. Design/methodology/approach: The paper begins with an introduction to corporate sustainability, an outline of the European property market and of the drivers for, and challenges to, sustainability for property companies and a review of the characteristics of materiality and external assurance. The information on which the paper is based is drawn from the leading European commercial property companies’ corporate websites. Findings: The paper reveals that all of Europe’s leading property companies had either reported or provided information on sustainability but that only approximately half of these companies had embraced materiality or commissioned some form of independent external assurance as an integral part of their sustainability reporting processes. In many ways, this reduces the reliability and credibility of the leading property companies’ sustainability reports. Looking to the future, growing stakeholder pressure may force more of the leading European property companies to embrace materiality and commission external assurance as systematic and integral elements in the sustainability reporting process. Originality/value: The paper provides an accessible review of the current status of materiality and external assurance among Europe’s leading commercial property companies’ sustainability reporting and as such it will interest professionals, practitioners, academics and students interested in the sustainability in the property industry. © 2016, Emerald Group Publishing Limited.

AUTHOR KEYWORDS: Corporate sustainability;  Europe;  External assurance;  Materiality;  Property companies;  Real estate

DOCUMENT TYPE: Article

PUBLICATION STAGE: Final

SOURCE: Scopus

Chandan, H.C.

56111013900;

A Comparison of corporate sustainability reporting in Europe and the mena region

(2015) Comparative Economics and Regional Development in Turkey, pp. 230-251.

https://www.scopus.com/inward/record.uri?eid=2-s2.0-84958114681&doi=10.4018%2f978-1-4666-8729-5.ch010&partnerID=40&md5=23ca1d079fd2953c799a2b5b684a68a8

DOI: 10.4018/978-1-4666-8729-5.ch010

AFFILIATIONS: Argosy University, United States

ABSTRACT: Corporate sustainability (CS) is becoming a strategic focus for large corporations globally. This chapter compares CS reporting in the Europe and MENA regions. A content analysis of the CS reports from large corporations in Europe and the MENA region is presented to explore the themes covered in their sustainability reports. More large corporations in Europe have been reporting CS longer than those in the MENA region, and more large corporations in the developed countries of Europe as compared to MENA region publish CS reports. In general, CS reporting in the MENA region and developing European countries is in its infancy. There is evidence of leadership structures being put in place to ensure that the board and senior management are involved in sustainable strategy development and are incentivized to monitor and ensure implementation of that strategy through financial rewards. © 2016 by IGI Global. All rights reserved.

DOCUMENT TYPE: Book Chapter

PUBLICATION STAGE: Final

SOURCE: Scopus

Papenfuß, U.

22734552500;

How (should) public authorities report on state-owned enterprises for financial sustainability and cutback management-a new quality model

(2014) Public Money and Management, 34 (2), pp. 115-122. Cited 7 times.

https://www.scopus.com/inward/record.uri?eid=2-s2.0-84893531614&doi=10.1080%2f09540962.2014.887519&partnerID=40&md5=fd55f736306b0ecc5dd1e955fe8417dc

DOI: 10.1080/09540962.2014.887519

AFFILIATIONS: University of Leipzig, Germany

ABSTRACT: Empirical data show the significance of state-owned enterprises (SOEs) for providing public services. Financial sustainability, cutback management and budget consolidation are no longer possible without including SOEs. This paper examines the ways that public authorities reported on the capital, performance and debts of their SOEs in Germany, Austria and Switzerland between 2009 and 2012. The quality of holdings reporting was found to differ quite considerably. The author provides new knowledge and a conceptual approach for countries all over the world to evaluate and substantially enhance public management concerning financial sustainability and cutbacks. © 2014 © 2014 CIPFA.

AUTHOR KEYWORDS: Accountability;  cutback management;  financial sustainability;  public corporation

DOCUMENT TYPE: Article

PUBLICATION STAGE: Final

SOURCE: Scopus

Steurer, R., Konrad, A.

6508332586;25632611400;

Business-society relations in Central-Eastern and Western Europe: How those who lead in sustainability reporting bridge the gap in corporate (social) responsibility

(2009) Scandinavian Journal of Management, 25 (1), pp. 23-36. Cited 54 times.

https://www.scopus.com/inward/record.uri?eid=2-s2.0-60949087941&doi=10.1016%2fj.scaman.2008.11.001&partnerID=40&md5=e4a445f5d68568dfbbb0bef2cbb0ff3f

DOI: 10.1016/j.scaman.2008.11.001

AFFILIATIONS: Institute of Forest, Environmental and Natural Resource Policy, University of Natural Resources and Applied Life Sciences, Vienna, Austria;

RIMAS - Research Institute for Managing Sustainability, Vienna University of Economics and Business Administration, Austria

ABSTRACT: In Western Europe, corporate (social) responsibility (CR) has become a popular concept that no major company can afford to ignore. However, what about the major companies from the new Central-Eastern Europe (CEE) Member States? The present paper is one of the first attempts to analyse the understanding and relevance of the CR of some major CEE companies that are leaders in sustainability reporting. This analysis is conducted in direct comparison with a similar analysis on major Western European companies. Methodologically, the paper intertwines two qualitative strands of research: an analysis of 19 CR reports (12 from CEE and 7 from Western Europe) provides a general impression about the understanding of CR across different socio-political contexts. This report-based depiction is complemented by two surveys of 22 companies (11 from CEE and 11 from Western Europe). The surveys show the relevance that the companies attach to specific CR issues. Overall, the paper concludes that the understanding of CR is context-specific, but also that, in the case of major companies that are leading in CR reporting, the differences are not as stark as one might expect. © 2008 Elsevier Ltd. All rights reserved.

AUTHOR KEYWORDS: Central-Eastern Europe (CEE);  Corporate (social) responsibility (CR);  Corporate sustainability;  Eastern Europe;  Environmental reporting;  Global Reporting Initiative/GRI;  Stakeholder management;  Sustainability reporting;  Sustainable development;  Western Europe

DOCUMENT TYPE: Article

PUBLICATION STAGE: Final

SOURCE: Scopus

Salzmann, O., Ionescu-Somers, A., Steger, U.

16204029600;6507271877;56528140400;

Corporate sustainability as an indicator for more humanism in business? A view beyond the usual hype in Europe

(2009) Humanism in Business, pp. 299-308.

https://www.scopus.com/inward/record.uri?eid=2-s2.0-84926102088&doi=10.1017%2fCBO9780511808395.019&partnerID=40&md5=2987b3176f7a87e18ff60bf001a66513

DOI: 10.1017/CBO9780511808395.019

AFFILIATIONS: IMD, Lausanne, Switzerland

ABSTRACT: The terms “corporate social responsibility” and “corporate sustainability” are used (often interchangeably) to describe the situation when companies take on a wider array of social and environmental issues (directly or indirectly) associated with their activities and take actions to mitigate those issues. A so-called business case for sustainability exists if these actions also translate into improved corporate financial performance. Companies in an ideal humanistic business environment would proactively decide in favor of actions that put society's needs first. In any case, a sound business case for corporate sustainability is an important foundation for the necessary attitudinal change in this regard – and a simple fact of making smart investments. Humanism can be interpreted as a more far-reaching version of corporate sustainability, one that most hard-nosed and myopic managers would most likely disregard as philanthropy. Sustainability in today's business environment has become increasingly prominent over the years, most recently in the context of climate change: businesses, policy-makers, and NGOs are attempting to surpass each other through – more or less – well-designed initiatives. And the media are happy to feed the heightened societal awareness. However, how much of this is hype and opportunism? How significant is stakeholder interest in corporate sustainability, as a first step towards achieving humanism in business? To what extent do they reward corporate leaders and penalize laggards? We hope to provide solid answers on the following pages, and will base them on empirical evidence collected at the International Institute for Management Development. © Cambridge University Press 2009.

DOCUMENT TYPE: Book Chapter

PUBLICATION STAGE: Final

SOURCE: Scopus

Author Response

Dear Reviewer,

Thank you for your valuable comments on the circular economic concept and on improving the quality of life. Additional references have been added to the introductory section to reinforce the context of the text. The additions are marked in red in the introductory chapter.

Comment: „a) The formulation of Tables and Figures has been bounded on indices and factors’ notation, while it is not straightforward which services/products/physical ontologies are derived from the “state-owned companies” mentioned? These companies sustain, probably, distinct features to each other, thus, diversified corporate sustainability “footprint” is anticipated. Therefore, a list of such typical ontologies/products served (no brand names) can be given in an extra Table.”

Answer: The research contract does not allow the listing of activities in an extra table or the publication of the results of the study in more detail, and therefore the companies and company profiles involved are not named. (Due to the monopolisation of certain public services, state-owned companies can be easily identified.)

Thank you for your comments and comments on corporate sustainability. Related articles were very helpful. These were read and processed. Those that were able to fit into the logical structure of the article (4 references) can be found in the reference list. Special thanks for the critical comments on corporate sustainability, which were really helpful comments. The corrections made are marked in red in the text and can be found in the body.

We have read each article, of which we have inserted the following:

Taliento, M., Favino, C., Netti, A. Impact of environmental, social, and  governance  information on economic performance: Evidence of a corporate 'sustainability advantage' from Europe(2019) Sustainability (Switzerland), 11 (6), art. no. 1738

Aravossis, K.G., Kapsalis, V.C.,Kyriakopoulos, G.L., Xouleis. Development of a holistic assessment framework for industrial organizations (2019) Sustainability (Switzerland), 11 (14), art. no. 3946

Kapsalis, V.C., Kyriakopoulos, G.L., Aravossis, K.G. Investigation of ecosystem services and circular economy interactions under an inter-organizational framework (2019) Energies, 12 (9), art. no. 1734

Ntanos, S., Kyriakopoulos, G., Chalikias, M., Arabatzis, G., Skordoulis, M., Galatsidas, S., Drosos, D. A social assessment of the usage of Renewable Energy Sources and its contribution to life quality: The case of an Attica Urban area in Greece(2018) Sustainability (Switzerland), 10 (5), art. no. 1414